

# Assessment of upper limb motor control: establishing normative benchmarks for clinical applications

Pablo Martín-Sierra, Cristina Sanchez, Eloy Urendes and Rafael Raya

Information Technologies Department, Universidad de San Pablo CEU, Boadilla del Monte, Madrid, Spain

Corresponding author
Pablo Martín-Sierra,
p.martin61@usp.ceu.es

## ABSTRACT

**Background**. This study presents and validates a methodology for assessing upper limb motor control using quantitative metrics derived from biomechanics and surface electromyography (sEMG). By combining kinematic and neurophysiological measurements, the study establishes normative benchmarks in healthy adults, which can offer a foundation for future clinical applications in populations with neuromotor impairments, which are commonly characterized by spasticity, involuntary coactivation, and restricted range of motion (ROM). A combination of clinical functional scales with the quantitative metrics presented here is expected to enable better evaluation of motor control.

**Materials and Methods**. Twenty healthy adults performed elbow flexion-extension (FE) movements at three controlled speeds (42, 60, and 78 beats per minute (bpm)). The protocol involved recording sEMG signals of the biceps brachii and triceps brachii, with an inertial measurement unit (IMU) sensor, to compute four metrics: the range of motion (ROM), a derived angular velocity, the coactivation coefficient (CC), and muscle synergy. Movements were segmented into acceleration and deceleration phases to enable a phase-specific analysis, with a focus on both agonist and antagonist muscle activity during flexion and extension.

**Results**. The results established normative values for each metric, showing alignment with previous results in the literature for healthy motor patterns. ROM values were consistent with the expected ranges for healthy adults ranging between normative values, with the angular velocity increasing proportionally to the speed of movement but showing greater variability at higher speeds. The CC analysis demonstrated distinct phase-dependent activation patterns, with higher values during flexion deceleration due to antagonist muscle stabilization requirements. The muscle synergy metric highlighted a balanced activation of the biceps and triceps, with minor secondary activation of the triceps during flexion to counteract gravitational forces.

**Discussion**. The results validate the feasibility of this approach for quantifying motor control based on the quantitative metrics presented here. Normative values and the ability to detect changes in ROM, CC, and muscle synergy enhance the diagnostic potential of this approach in terms of identifying spasticity, coordination deficits, or abnormal neuromuscular patterns in clinical populations. This study establishes a comprehensive methodology for evaluating upper limb motor control, based on a combination of kinematic and neurophysiological data. These findings offer a solid foundation for developing advanced diagnostic tools and personalized rehabilitation strategies, with potential applications to conditions such as stroke, cerebral palsy, and other neuromotor impairments.

# INTRODUCTION

Objective assessment of motor control is essential in order to develop personalized and effective rehabilitation strategies, particularly for people with neuromotor disorders such as stroke or cerebral palsy (CP). People with these disorders suffer from motor impairments including muscle spasticity, muscle weakness, abnormal range of motion (ROM), and problems with balance and coordination. These impairments hinder everyday tasks, with spasticity posing particular challenges by hindering voluntary movement and limiting mobility, and often leading to joint deformities over time (*Ötensjø, Carlberg & Vøllestad, 2004*)

Traditional clinical functional scales such as the Quality of Upper Extremity Skills Test (QUEST) (*Thorley et al., 2012*) and the Manual Ability Rating System (MACS) (*Eliasson et al., 2006*) are widely used to assess motor function, whereas spasticity is often assessed using the Modified Ashworth Scale (MAS) (*Brotnick, 2023*). However, these methods involve a degree of subjective interpretation, which could influence the accuracy needed to monitor progress or evaluate therapeutic interventions.

Recent research has highlighted the potential of quantitative metrics derived from kinematic and neurophysiological data, such as ROM and surface electromyography (sEMG), respectively, to address these limitations and complement traditional clinical scales (*He et al., 2023*). ROM is a widely used measure for the assessment of joint flexibility, and determines the extent of movement a joint can achieve within its functional limits. Angular velocity, a metric derived from the ROM, quantifies the speed and smoothness of movement cycles, and provides additional insights into motor control and coordination. From a neurophysiological perspective, the coactivation coefficient (CC) evaluates the degree of simultaneous activation of antagonist muscles (*Maura et al., 2023*; *Wu et al., 2024*), while muscle synergy analysis examines the balance of activation between antagonist and agonist muscles across different phases of movement (*Le Bozec, Maton & Cnockaert, 1980*). These metrics provide insight into alterations in neuromuscular control, such as altered firing sequences or excessive coactivation, which may be associated with spasticity (*Falconer & Winter, 1985*; *Sarcher et al., 2017*). Together, ROM and sEMG analyses provide a comprehensive assessment of the mechanical and neural aspects of motor function, which can contribute to the diagnosis, monitoring and treatment of these conditions (*Campanini et al., 2020*).

This study introduces and evaluates a methodology for analysing motor state by combining kinematic measurements, including ROM and angular velocity, with neurological metrics such as CC and muscle synergy. These measurements are derived from flexion-extension (FE) movements of the elbow, recorded using a protocol aligned with previous work by *Sarcher et al. (2015)* and *Sarcher et al. (2017)*. Data from healthy adults will establish normative values, providing a benchmark for future studies.

Specifically, ROM in healthy individuals is assessed in order to compare the obtained values with normative data available in the literature and to establish updated values for future studies. A similar methodology is applied to angular velocity; however, in the absence of existing normative data for this parameter, the findings provide novel insights into its role in motor control. Given that angular velocity is derived from ROM, and that our ROM values are aligned with established standards, it is plausible to consider the angular velocity measurements presented here as representative of normative ranges.

Furthermore, metrics based on sEMG are analyzed to evaluate the presence and extent of muscle coactivation during elbow FE movements in a normative population. Muscle coactivation has been posited in previous studies as essential for effective motor control (*Hogan, 1984*; *Falconer & Winter, 1985*). In addition, CC and muscle synergy metrics are compared with prior findings (*Le Bozec, Maton & Cnockaert, 1980*), with the aim of contributing to the refinement and expansion of normative data for these parameters.

The normative values derived from this study will ultimately support the assessment of motor impairments in clinical populations, such as people with CP, thereby facilitating more effective diagnosis, monitoring and treatment strategies. The contributions of this paper must therefore be considered as a requisite step before applying these proposed metrics to the evaluation of motor control of people with neuromotor disorders. Therefore, the study hypothesizes that significant differences will be found in the measured metrics across different movement phases. Since the study population consists of healthy individuals, we expect a high level of homogeniety between subjects.

# MATERIALS AND METHODS

The overall workflow of the research process is outlined in Fig. 1. This flowchart provides a high-level overview of the steps involved, which are then detailed in the following subsections. Each subsection describes the specific methods and procedures used in this study.

## Participants and protocol

The study involved 20 healthy adults, aged 29.69 $\pm$ 8.01 years, all with normative motor development and no known motor or cognitive impairments. The sample size employed in this study was selected as an initial step to evaluate the feasibility and reliability of a novel methodology and associated metrics for the assessment of motor control during elbow flexion-extension. Although a group of 20 healthy participants does not permit the establishment of definitive normative values, the primary aim was to determine whether the proposed approach yields stable and reproducible measurements under controlled conditions. These preliminary data are intended to serve as a foundation for subsequent investigations involving larger and more heterogeneous populations, including clinical cohorts with motor impairments. Despite its limited size, the selected sample was sufficient to produce consistent group-level trends and to facilitate a focused validation of the experimental protocol (*Brookshaw, Sexton & McGibbon, 2020*; *Hernández et al., 2023*).

Each participant performed elbow FE movements with the dominant arm, performed in a neutral pronosupination position and following an external rhythm set by a metronome

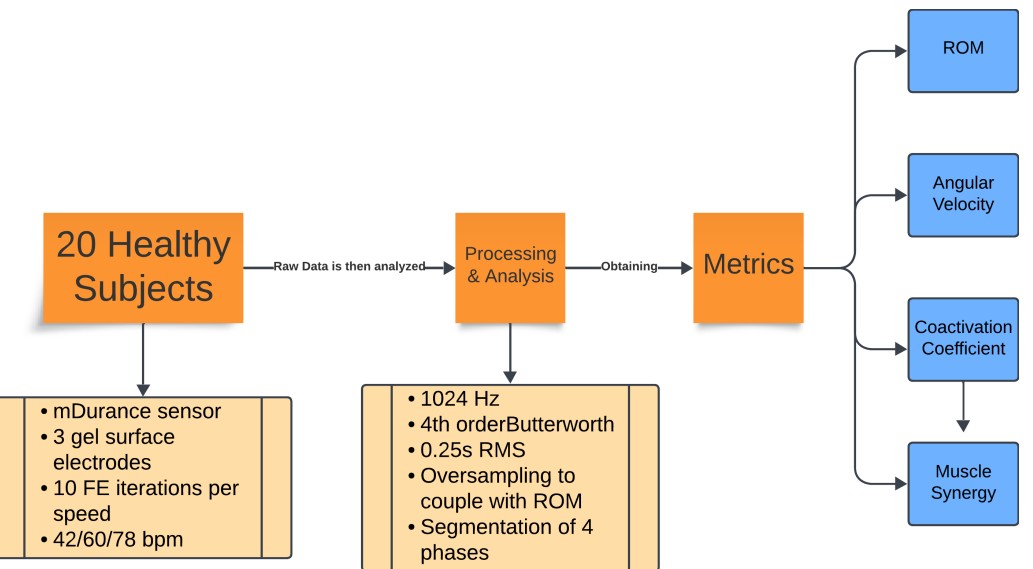

**Figure 1** **Flowchart of the experiment methodology.** Summary of the experimental protocol and data analysis. Twenty healthy subjects performed 10 repetitions of flexion-extension at three cadences (42, 60, and 78 bpm). Electromyography (EMG) data were recorded with mDurance sensors, sampled at 1,024 Hz, filtered, and processed using RMS and oversampling to align them with the range of motion data. The movements were divided into four phases, from which the range of motion, angular velocity, coactivation coefficient, and muscle synergy metrics were extracted.

at three speeds (42, 60, and 78 beats per minute (bpm)), which were chosen to mimic experimental conditions from the literature (*Sarcher et al., 2015*; *Sarcher et al., 2017*). A metronome can also aid in minimizing trunk displacement during the exercise, according to prior studies (*Van Roon, Steenbergen & Meulenbroek, 2005*). The FE movement features a single degree of freedom, thus minimizing noise from other joints such as the wrist. It also involves a clearly defined pair of agonist and antagonist muscles, the biceps brachii and triceps brachii, from which sEMG signals were recorded as participants performed the protocol. Finally, FE movement is essential in performing daily activity tasks, making it an optimal candidate in view of its functional contribution. Participants were instructed to execute 10 repetitions per speed, resulting in a total of 30 FE cycles. Two-minute pauses were taken between each set of recordings, in order to adjust the metronome and to prevent muscle fatigue to the participants. To maintain consistency, participants were seated with their back supported and straight, with knees at 90° to the floor, arms free of additional weight or resistance, and elbows aligned with the plane of the sEMG+IMU sensor.

In addition, participants were asked to perform the maximum possible ROM without reaching hyperflexion or hyperextension, starting and ending the movement in the maximum flexion position. Both wrist and shoulder joints were to remain as static as possible, so that no compensatory movements were performed which could affect the recorded measurements.
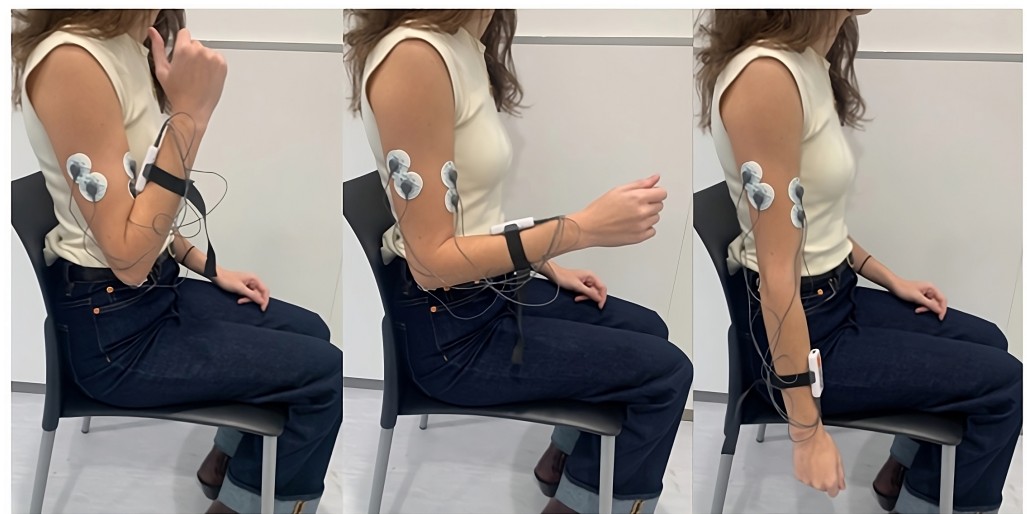

**Figure 2 Electrode placement for sEMG recording on the biceps brachii and triceps brachii muscles, following SENIAM guidelines.** The image also shows an example of the FE cycle motion used during the study protocol.

All participants signed an informed consent form prior to the protocol, and the protocol was approved by the Research Ethics Committee of the San Pablo CEU University (561/21/53).

## Instrumentation

sEMG signals and motion data were collected using an mDurance™ sensor (https://mdurance.com/en/), a wireless, dual-channel device employed to record signals from the biceps brachii and triceps brachii. This sensor has an integrated inertial measurement unit (IMU) for tracking the ROM in the plane of the movement. Each subject's arm was fitted with the sensor, which was positioned to align with the plane of elbow flexion and extension to ensure proper acquisition of the ROM.

Five pre-gelled surface electrodes (42x36 mm) were used, two for the biceps brachii, two for the triceps brachii longus, and one for the elbow bone (acting as a reference electrode), following the electrode placement guidelines recommended by the SENIAM (*SENIAM, 2024*). The positions of the electrodes and an example of the FE cycle motion, according to the prior protocol, can be seen in Fig. 2. These electrodes were connected to the mDurance™ control unit, which was placed on the subject's forearm with the help of an elastic strap. The sensor was placed in the centre of the forearm to coincide with the plane of the FE movement (sagittal plane), since this was important for the proper acquisition of the ROM.

Data were visualized in real time using the mobile app for the mDurance™ system, which included a built-in metronome to control the movement rhythm and a server interface to allow for the monitoring of data and identification of potential measurement errors.

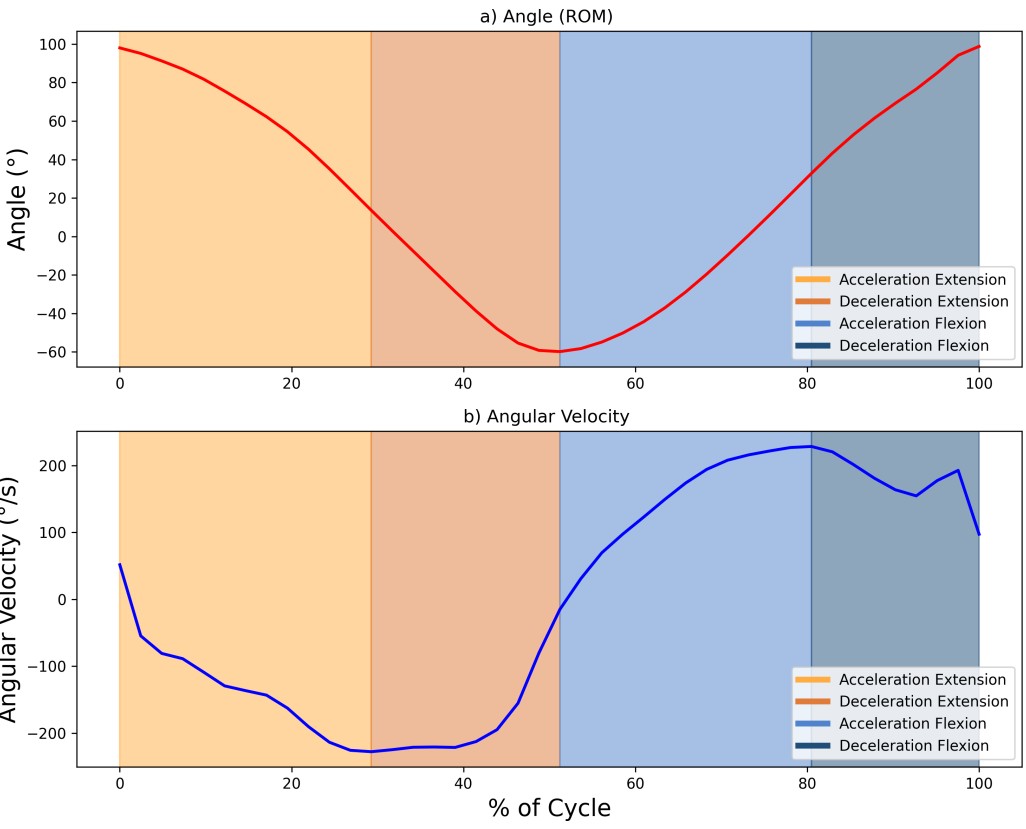

**Figure 3** **Example showing the diûerent phases of movement within one FE cycle at 60 bp.** (A) Angle (°) and (B) angular velocity (°/s), with changes in area colour indicating transitions between the acceleration and deceleration phases for flexion and extension. Maximum and minimum velocity and angle points mark the phase boundaries.

## Signal processing and data segmentation

The sEMG signals were sampled at 1,024 Hz and processed using a fourth-order Butterworth bandpass filter (20–450 Hz). These signals were then smoothed with a 0.25-s root mean square (RMS) sliding window, producing an envelope signal with a sampling rate of 4 Hz. To ensure alignment with the ROM data, which were sampled at 20 Hz, the envelope signal was oversampled. This alignment allowed for precise delineation of the phases of the FE cycle, using ROM peaks (with 0° indicating the arm's neutral position at a 90° elbow flexion) as reference points to segment and identify the phases of the movement. The maximum and minimum values of the ROM (varying between 75° and −60°) corresponded to the maximum flexion and extension values, respectively. Further segmentation of the motion into acceleration and deceleration phases was carried out based on the angular velocity, calculated as the time derivative of the ROM. Hence, each FE movement was divided into four segments (acceleration and deceleration phases for elbow flexion, and the same phases for elbow extension, resulting in a total of 40 segments per condition subject-speed). Figure 3 illustrates the segmentation of movement phases within a single FE cycle.

Baseline correction was used to properly align the sEMG envelope with the subject's muscle activity. Using 30 samples taken around the point of minimum activity, the mean and standard deviation were calculated, with the offset taken as the mean minus three standard deviations. This method, based on work by *Sarcher et al. (2015)* and *Sarcher et al. (2017)* ensured an accurate representation of the activity. To complete the pre-processing steps, the sEMG envelope was normalized to the maximum contraction peak recorded during each register (*Falconer & Winter, 1985*). The algorithm, including both the preprocessing steps and the calculation of the proposed metrics, was developed in Python 3 (Jupyter Notebook 6.4.5 environment).

## Quantitative metrics

To provide a comprehensive analysis of motor control, four metrics were calculated:

- **ROM and angular velocity**: the ROM is obtained as the total angular rotation between maximum flexion and maximum extension. This quantitative metric represents the maximum movement achievable and the extent of joint flexibility, and is essential for establishing normative benchmarks for the range of movement. The availability of studies in which normative ROM values have been defined for healthy individuals enables future comparisons with data obtained from individuals with motor control impairments (*Maura et al., 2023*; *He et al., 2023*; *Wu et al., 2024*). Following the measurement of the ROM, the angular velocity was computed from the ROM values. The standard deviation of the angular velocity at each time point can be analyzed to assess the consistency of execution of the movement. A higher standard deviation indicates greater variability in performance, which may reflect fluctuations in neuromuscular control or movement efficiency (*Wakeling, Blake & Chan, 2010*). The average ROM values, along with their standard deviations and the angular velocity values, were calculated and depicted for one average FE cycle representation with their standard deviation ranges across all points.

- **Coactivation coefficient (CC)**: The CC quantifies the simultaneous activation of antagonist muscles, specifically the biceps brachii and triceps brachii in elbow FE movements. Using the method of Falconer and Winter (*Falconer & Winter, 1985*), the CC was calculated as the common area under the sEMG envelope curve for both antagonist and agonist muscles during each FE cycle (see Eq. (1), where Ag is the area under the envelope of the agonist muscle, Antag is the area under the envelope of the antagonist muscle and (Ag, Antag) is the common area of both muscles).

$$CC = 2 \cdot \frac{\int (Ag, Antag)}{\int Ag + \int Antag} \cdot 100 \tag{1}$$

Low CC values, which indicate minimal activation of the antagonist relative to the agonist, are generally associated with efficient motor control and precise movements. High CC values indicate greater overlap in the activation of antagonistic and agonistic muscles; although this is necessary for joint stability in certain movement phases, it may suggest spasticity when excessively elevated (*Falconer & Winter, 1985*; *Sarcher et al., 2017*).

For this analysis, the CC was also computed separately for the acceleration and deceleration phases within both the flexion and extension movements. The separation of the

acceleration and deceleration phases is relevant, as these phases have distinct characteristics: the acceleration phase is associated with the generation of speed and momentum, whereas the deceleration phase is characterized by a focus on stability and control. Consequently, an increase in coactivation is expected toward the end of the movement, to facilitate stability and precision during its conclusion (*Sarcher et al., 2015*)

The choice of this method for calculating the CC, as opposed to other approaches described in the literature (*Hogan, 1984*), was made to avoid the need to obtain the maximum voluntary contraction (MVC). Instead of normalizing the EMG signal based on this reference, it was normalized, as stated above, to the maximum contraction peak recorded during each register. This method facilitates the future acquisition of the CC in individuals with motor impairments (*e.g.*, children with CP), where the assessment of MVC is challenging, primarily due to conditions such as hypotonia and spasticity (*Burden, 2010*).

A statistical analysis was conducted that included a two-way analysis of variance (ANOVA) to assess the effects of two factors, the phase of movement and the movement speed, on the mean CC values. The ANOVA was followed by a *post-hoc* analysis performed using Tukey's Honest Significant Difference (HSD) test to identify specific group differences within each factor (phase and speed). these significant results are highlighted in the tables in the Results section with * for $p < 0.05$, ** for $p < 0.01$, and *** for $p < 0.001$. Following this, Cohen's $f$ statistic was also calculated to assess the magnitude of the effect, and the value for the movement phase factor was 0.4123 (indicating a strong effect). Cohen's $f$ values of 0.10 are considered of small effect, those of 0.25 are of a medium effect, and higher than 0.40 are considered large (*Cohen, 1988*).

CC values are illustrated in the form of bar plots, with one bar for each movement phase and each movement speed, as well as one bar plot showing the average CC for all movement speeds combined.

• **Muscle synergy**: This is defined as the proportion of activation contributed by each muscle during the flexion and extension phases. It was calculated as the percentage of the sEMG envelope area for the biceps brachii (primary during flexion) and triceps brachii (primary during extension) over the course of each cycle of FE movements. In each FE cycle, the muscle activity can be divided into its agonist phase and its antagonist phase, and based on the muscle synergy, we can obtain the percentage of activity in one phase with respect to the other.

# RESULTS

The values of each metric were analyzed across all participants, for each movement speed (42, 60, and 78 bpm).

• **Range of motion and angular velocity**

All participants demonstrated ROM values that were consistent with normative data, with values ranging from approximately 75° to −60° for elbow FE (*Maura et al., 2023*; *He et al., 2023*; *Wu et al., 2024*). This broad ROM indicates unrestricted joint mobility, and verifies the existing reference range for ROM in a healthy population. Such data provide

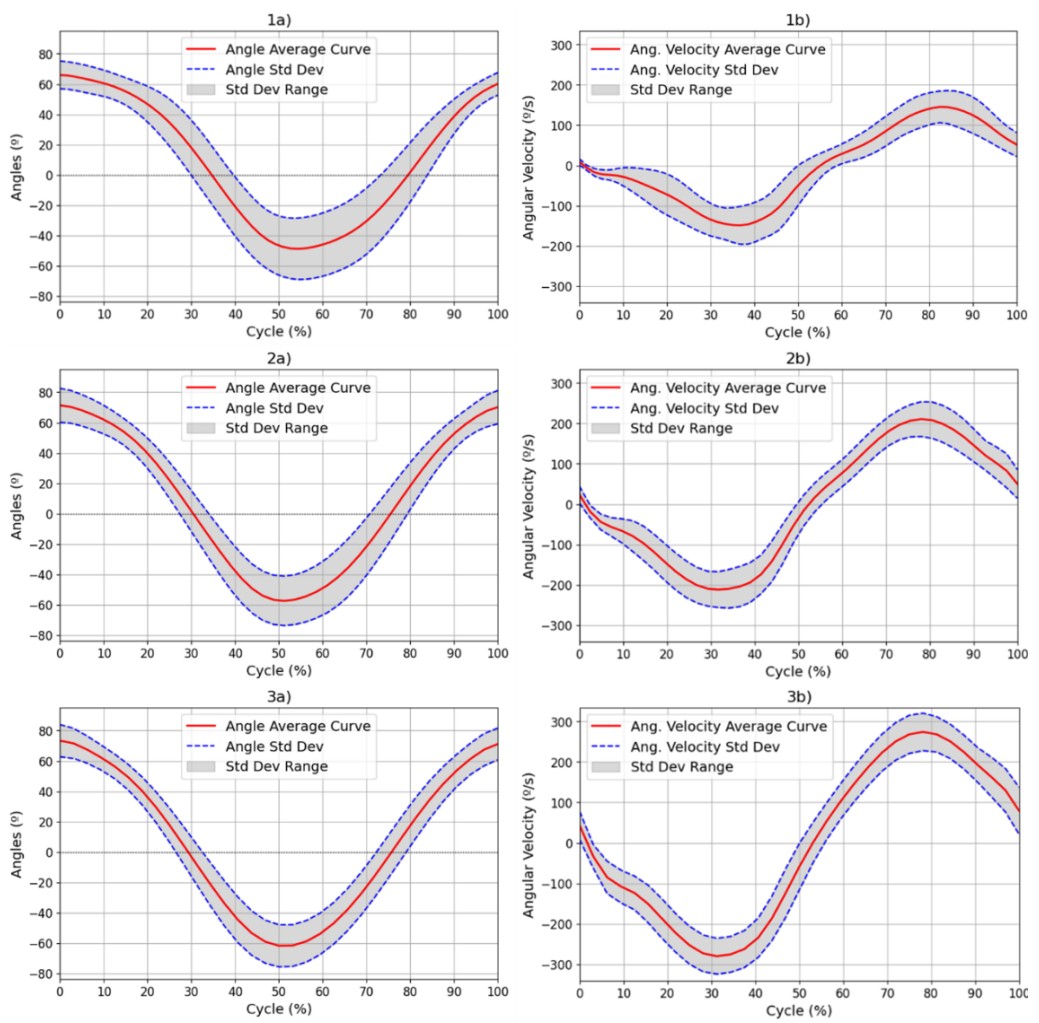

**Figure 4** Results for the (A) ROM and (B) angular velocity, with mean and SD values for all subjects for each of the movement speeds: 42 (1), 60 (2) and 78 (3) bpm. Results for the (a) ROM and (b) angular velocity, with mean and SD values for all subjects for each of the movement speeds: 42 (1), 60 (2) and 78 (3) bpm. The red line represents the mean values at each point of the cycle, while the blue dotted line represents the standard deviation at each point. The range of this standard deviation is shaded in grey.

essential baseline values for later comparison with patient populations. Figure 4 illustrates the average values and the standard deviation of the ROM and angular velocity, for all movement speeds (1–3) over all FE repetitions. Table 1 presents the numerical values obtained for the overall ROM and angular velocity values as well as the average standard deviation of the angular velocity values and the maximum and minimum values of the standard deviation in the angular velocity.

Although other kinematic metrics, such as the percentage of time taken to reach the maximum velocity in each phase of the movement or the maximum standard deviation in repetitions, could provide more information on the variability and dynamics of the
**Table 1   ROM and speed measurements for all subjects at each movement speed.**

| Freq. | 42 bpm | 60 bpm | 78 bpm |
|---|---|---|---|
| **ROM (°)** | [66.92, −50.67] | [72.46, −58.36] | [74.58, −62.82] |
| **Avg. Stdev. of ROM (°)** | 16.06 ± 5.62 | 13.54 ± 3.36 | 11.49 ± 2.15 |
| **Angular velocity (°/s)** | [160.24, −174.87] | [221.17, −233.49] | [283.95, −294.81] |
| **Avg. Stdev. of angular velocity (°/s)** | 37.69 ± 12.71 | 38.12 ± 8.47 | 43.47 ± 5.32 |

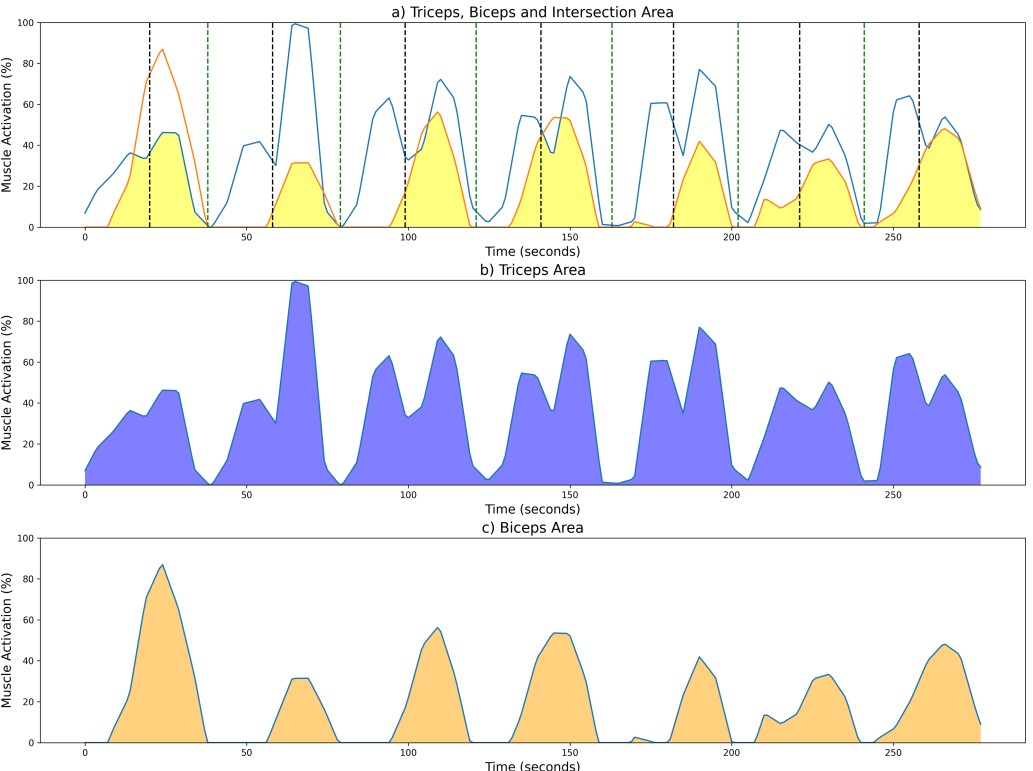

**Figure 5   Examples of CC results from a measurement session at 42 bpm for the unsegmented sEMG envelope.** (A) Common activation area between biceps (orange) and triceps (blue); individual activation areas for the (B) triceps and (C) biceps. Black and green dotted lines mark the maximum extension and fexion points, respectively.

movement, the current analysis focuses on characterizing ROM and angular velocity. As such, mean ± standard deviation was used as the primary summary measure in this study.

- **Coactivation coefficient**

To aid in the visualization of the CC computation of the unsegmented sEMG envelope signal, Fig. 5 illustrates three graphs. Figure 5A shows the shared area under the sEMG envelope signals for the biceps and triceps muscles. The black dotted lines denote the point of maximum extension, while the green dotted lines indicate the point of maximum flexion. Figures 5B and 5C show the entire activation area for the triceps and biceps, respectively.

As stated above, the CC was also analyzed for each of the four segments or phases of the FE cycle: flexion acceleration (Acc_Flex), flexion deceleration (Dec_Flex), extension

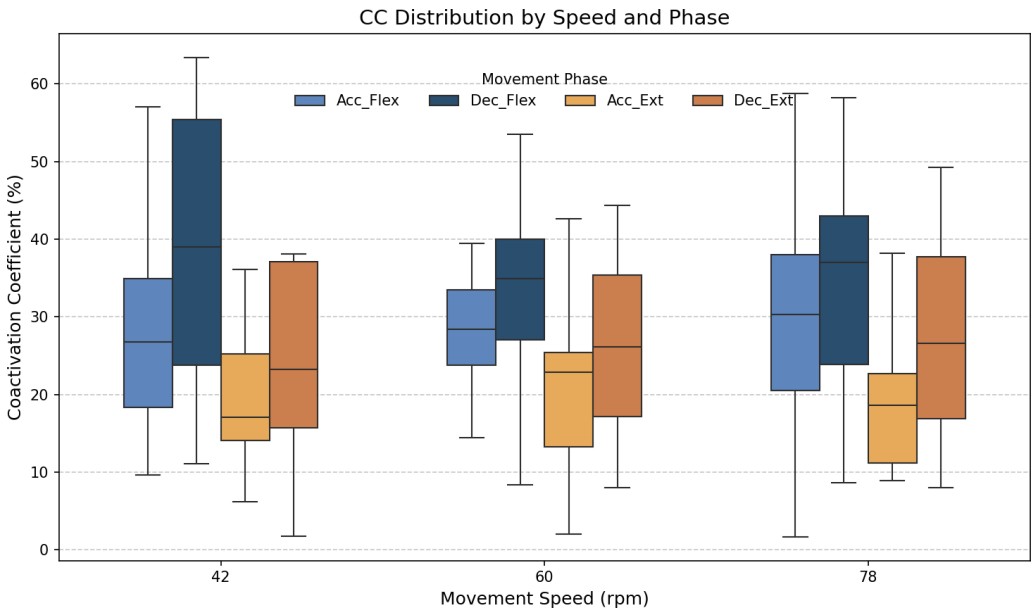

**Figure 6** **Box plots of the coactivation coeffcient values (mean ± SD) for the dominant arm, for the four different phases of movement.** The four different phases of movement are: acceleration of flexion (Acc_Flex), deceleration of flexion (Dec_Flex), acceleration of extension (Acc_Ext), and deceleration of extension (Dec_Ext).

acceleration (Acc_Ext), and extension deceleration (Dec_Ext). CC values were also computed for each of the three experimental speeds, yielding a mean and standard deviation across all FE cycles for all subjects (Fig. 6). The CC was higher during the flexion phase than the extension phase and increased proportionally with movement speed.

It was hypothesized that the CC would significantly differ across the phases of movement and potentially vary with speed. To test this hypothesis, a two-way ANOVA was conducted to evaluate the main effects of movement phase and speed, as well as their interaction, on CC values. The results are presented in Table 2. The results showed a significant main effect of movement phase ($F(3,188) = 12.84$, $p < 0.001$), but no significant effect of speed ($F(2,188) = 0.07$, $p = 0.93$), nor a significant interaction between phase and speed ($F(6,188) = 0.21$, $p = 0.97$). Given the absence of a significant speed effect, the data across all speeds were pooled to further investigate the influence of movement phase alone. The ANOVA was followed by a *post-hoc* analysis using Tukey's Honest Significant Difference (HSD) test, which includes a built-in correction for multiple comparisons to control the family-wise error rate (FWER). This test was selected because it is well-suited for all pairwise comparisons while maintaining the overall significance level at $\alpha = 0.05$. The results for the factors of movement phase and speed are presented in Tables 3 and 4, respectively. Significant differences were found only between the groups of the movement phase factor, with no significant differences in the speed factor. All pairwise comparisons, except for Acceleration_Flexion and Deceleration_Extension, had *p*-values below 0.05.
**Table 2 ANOVA table values.** Differences between groups were considered statistically significant at $p < 0.05$ and are indicated by asterisks (* for $p < 0.05$, ** for $p < 0.01$, and *** for $p < 0.001$).

| Source | SS | df | F-value | p-value |
|---|---|---|---|---|
| **Phase** | 7,336.39 | 3 | 12.84 | <0.001*** |
| **Speed** | 26.54 | 2 | 0.07 | 0.93 |
| **Phase × Speed** | 242.10 | 6 | 0.21 | 0.97 |
| **Residual error** | 35,816.19 | 188 | | |

**Table 3 Tukey p-values for movement phase.** Differences between groups were considered statistically significant at $p < 0.05$ and are indicated by asterisks (* for $p < 0.05$, ** for $p < 0.01$, and *** for $p < 0.001$).

| Movement phase pairs | p-value |
|---|---|
| [Acc_Flex–Dec_Flex] | 0.0097** |
| [Acc_Flex–Acc_Ext] | 0.0102* |
| [Acc_Flex–Dec_Ext] | 0.9999 |
| [Dec_Flex–Acc_Ext] | 0.0000*** |
| [Dec_Flex–Dec_Ext] | 0.0079** |
| [Acc_Ext–Dec_Ext] | 0.0124* |

**Table 4 Tukey p-values for speed.**

| Angular velocity pairs | P-value |
|---|---|
| [42–60] bpm | 0.9490 |
| [42–78] bpm | 0.9602 |
| [60–78] bpm | 0.9996 |

Figure 7 shows the average values of CC for each movement across all speeds, which are as follows: 28.07% ± 11.22% for acceleration of flexion, 36.66% ± 15.52% for deceleration of flexion, 19.54% ± 9.77% for acceleration of extension and 27.9% ± 16.55% for deceleration of extension. As no statistically significant differences were found between speeds, all data was combined to obtain an overall coactivation patterns. An elevated value of CC during flexion can be seen just after the agonist (biceps) reaches its maximum activation, reflecting the additional triceps activation required to counter gravitational force and to ensure controlled movement6 shows the average values of CC for each movement across all speeds, which are as follows: 28.07% ± 11.22% for acceleration of flexion, 36.66% ± 15.52% for deceleration of flexion, 19.54% ± 9.77% for acceleration of extension and 27.9% ± 16.55% for deceleration of extension. An elevated value of CC during flexion can be seen just after the agonist (biceps) reaches its maximum activation, reflecting the additional triceps activation required to counter gravitational force and to ensure controlled movement.

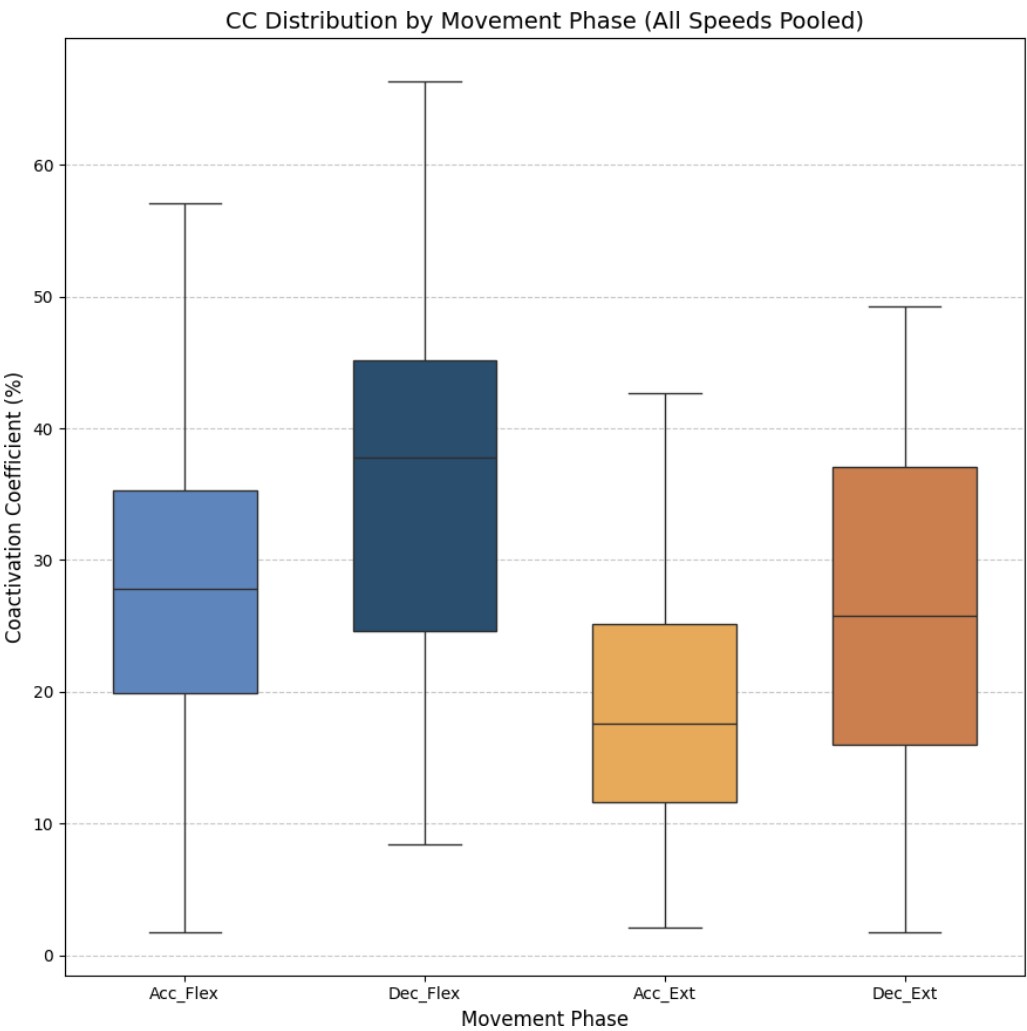

**Figure 7** **Box plots of the coactivation coeffcient values (mean ± SD) for combined speeds, for the four different phases of movement.** The four different phases of movement are: acceleration of flexion (Acc_Flex), deceleration of flexion (Dec_Flex), acceleration of extension (Acc_Ext), and deceleration of extension (Dec_Ext).

● **Muscle synergy**

As stated above, muscle synergy was evaluated by calculating the percentage of biceps and triceps activation during each FE cycle phase. The values obtained for all three movement speeds are illustrated in Fig. 8.

An analysis of muscle synergy for all participants demonstrated balanced activation between the biceps (predominantly active during flexion) and triceps (predominantly active during extension). Notably, the triceps was found to have increased activation during its antagonist phase (flexion), a phenomenon that was less pronounced in the antagonist phase of the biceps (extension).

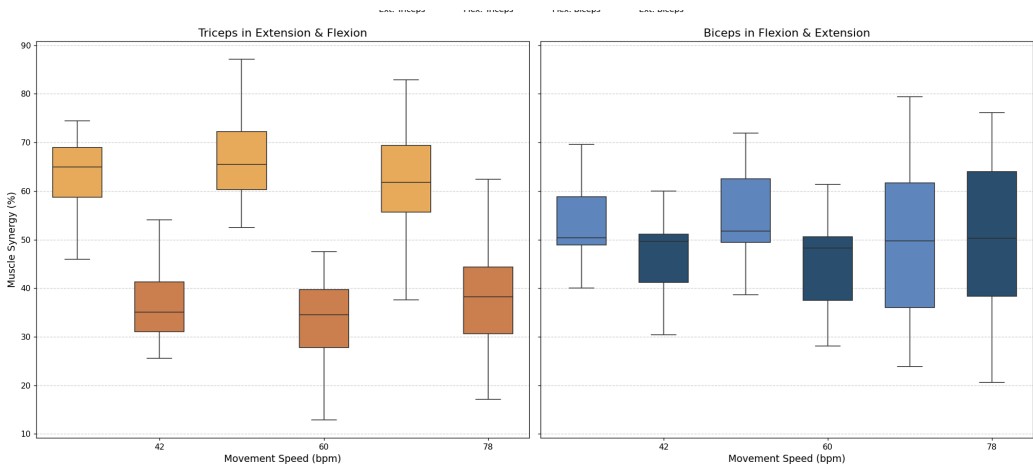

**Figure 8** Box plots of muscle synergy of the dominant arm of the antagonistic pair (triceps and biceps) in both the fexion and extension phases.

## DISCUSSION

The results of this study provide a detailed, quantitative characterisation of motor control during elbow FE movements in healthy adults and offer insights into the biomechanical and neurological aspects of the movement. By analysing the ROM, its derived angular velocity, CC and muscle synergy, this work establishes normative benchmarks that will be essential for future applications in clinical settings.

The ROM values observed in the study ranged from approximately −66.92° to 74.58° across the three speeds of movement (42, 60 and 78 bpm), as illustrated in Fig. 4 and Table 1. These results confirm that the experimental protocol effectively captures normal joint mobility in healthy individuals, as these values are within the normative range established by previous authors (*Zwerus et al., 2019*). Furthermore, angular velocity values increased proportionally with the speed of movement, and their standard deviations increased, reflecting less consistency and control of movement at higher speeds. Table 1 demonstrates that the mean standard deviation of the velocity increased from 37.69°/s at 42 bpm to 43.47°/s at 78 bpm. An increase in the minimum standard deviation of angular velocity from 5.64°/s to 23.6°/s indicated greater variability in movement as the speed increased. This increase can be explained by Fitts' Law, which indicates that the speed of a movement decreases the accuracy relative to a reference trajectory (*Fitts, 1992*; *Bucchieri et al., 2024*).

The CC results revealed different patterns of muscle coordination, emphasizing the role of antagonist activation in movement stabilization. Moreover, our results indicate the importance of analyzing the phase of movement (acceleration/deceleration and flexion/extension). These CC results are comparable to those obtained in previous work by *Sarcher et al. (2015)* and *Sarcher et al. (2017)*, although only extension CC values were measured in that study. The CC values are also comparable to those measured by *Falconer & Winter (1985)*. Higher CC values were observed during flexion than extension, particularly
in the deceleration phase, where the need for controlled braking against gravitational forces becomes prominent. As illustrated in Fig. 6, the CC values ranged from 27.09% ± 7.84% during flexion acceleration at 42 bpm to 38.8% ± 12.83% during flexion deceleration at the same speed. The ANOVA analysis showed significant differences in CC values between the phases of movement ($p < 0.05$), although no significant differences were found between speeds. A Tukey's *post hoc* analysis further revealed significant differences between all phases of movement except for flexion acceleration and extension deceleration ($p > 0.9999$). Cohen's $f$ statistic for the movement phase factor (0.4123) indicated a large effect size, thereby further validating the importance of phase-based differences in CC. Since more pronounced coactivation of the triceps, or its occurrence in a location different from that described above, may be associated with the presence of spasticity rather than with a movement control mechanism (*Li et al., 2019*), this coactivation effect was quantified in healthy subjects to offer insight into motor control in spasticity, and to allow for comparisons between normative and impaired patterns.

A muscle synergy analysis highlighted balanced activation patterns between the biceps and triceps across all phases of movement. Figure 8 shows these interactions, with primary biceps activity seen during flexion and primary triceps activity during extension. The primary activity of each muscle was observed during its corresponding agonistic phase. However, some activation of the muscles during their antagonistic phases was measured, more notably in the triceps. This observed increase in triceps activation during flexion can be attributed to a minor secondary activation, which is consistent with the gravitational counteraction mechanism as described in the analysis of the CC and depicted in Fig. 8. The secondary activation of the triceps during flexion further supports its role in counteracting gravitational forces, thus ensuring smooth and coordinated transitions between phases. These findings underscore the presence of well-coordinated neuromuscular activation in healthy individuals (*Hogan, 1984*; *Falconer & Winter, 1985*).

## CONCLUSION

This study investigated motor control during elbow FE movements in healthy adults, aiming to establish normative biomechanical and neurological benchmarks. By analyzing ROM, angular velocity, CC, and muscle synergy, the findings provide a comprehensive understanding of joint mobility, movement variability, and neuromuscular coordination.

The results confirm that the experimental protocol effectively captures normal joint mobility, with ROM values aligning with previously established norms. Angular velocity increased with movement speed, but with greater variability at higher speeds, indicating reduced control. CC analysis highlighted phase-dependent differences in muscle coordination, with antagonist activation playing a crucial role in movement stabilization, particularly during flexion deceleration. Muscle synergy analysis showed well-balanced activation between the biceps and triceps, ensuring smooth phase transitions. These findings contribute to existing knowledge by offering a multidimensional approach to motor assessment, combining biomechanical and neurological metrics.

In this regard, one notable finding is the level of coactivation between the biceps and triceps observed during the deceleration phase. While this specific coactivation has not

been extensively measured in previous studies, our results suggest that it may be essential for stabilizing the joint during this phase of movement. Although existing research, such as the studies from the 1980s (*Falconer & Winter, 1985*), has explored muscle coordination broadly, it has not directly quantified this interaction during deceleration. This finding could contribute to the broader understanding of muscle dynamics, particularly in movements that require joint stabilization.

The integration of these metrics provides a comprehensive framework for assessing motor control with a multidimensional view of motor performance. This allows for detailed assessments that go beyond traditional qualitative assessments. The results indicate strong potential for clinical translation. For individuals with neuromotor impairments, deviations in ROM, excessive CC values (especially in certain phases of movement), or unbalanced muscle synergy could serve as indicators of spasticity or impaired motor coordination.

Further studies will be needed to validate these findings in populations with motor impairments, including those with CP, stroke and similar neuromotor disorders. Expanding the dataset to include different age groups would improve the applicability of the benchmarks established here. The small sample size considered in the study and the narrow demographic focus may limit the generalizability of this framework; however, the detailed methodology of the study and analysis offers new insights into motor evaluation and suggests potential applications in rehabilitation.

While this study focused on single-plane flexion-extension movements to establish foundational benchmarks, future research should expand to include multi-axis, rotational movements to better simulate real-world functional tasks. These movements are clinically relevant, especially for conditions like stroke or cerebral palsy, where coordination across multiple joints and axes is often impaired. Including rotational and multi-axis dynamics will enhance the applicability of these normative values in clinical assessments and rehabilitation strategies.

In summary, this study has demonstrated the feasibility and utility of combining biomechanical and neurological metrics for objective assessment of motor control. The results provide a solid basis for the development of innovative diagnostic tools and personalized rehabilitation strategies, thereby advancing the field of neuromotor assessment and treatment.

## ACKNOWLEDGEMENTS

The authors thank the participants for their collaboration in the experiments.

### Funding

This experiment formed part of the NEUROMETRICS project, funded by MCIN/AEI/ 10.13039/501100011033/ERDF, EU, grant number: PID2021-127096OB-100. The funders had no role in study design, data collection and analysis, decision to publish, or preparation of the manuscript.

## Grant Disclosures

The following grant information was disclosed by the authors:
MCIN/AEI/10.13039/501100011033/ERDF, EU: PID2021-127096OB-100.

## Competing Interests

The authors declare there are no competing interests.

## Author Contributions

- Pablo Martín-Sierra conceived and designed the experiments, performed the experiments, analyzed the data, prepared figures and/or tables, and approved the final draft.
- Cristina Sanchez conceived and designed the experiments, authored or reviewed drafts of the article, and approved the final draft.
- Eloy Urendes conceived and designed the experiments, authored or reviewed drafts of the article, and approved the final draft.
- Rafael Raya conceived and designed the experiments, authored or reviewed drafts of the article, and approved the final draft.

## Human Ethics

The following information was supplied relating to ethical approvals (*i.e.*, approving body and any reference numbers):

Ethical approval was obtained from the Research Ethics Committee of the San Pablo CEU University (561/21/53).

## Data Availability

Raw data is available in the Supplemental Files.

## Supplemental Information

Supplemental information for this article can be found online at http://dx.doi.org/10.7717/peerj.19859#supplemental-information.

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
