# Peer review of "Assessment of upper limb motor control: establishing normative benchmarks for clinical applications"

_PeerJ, doi:10.7717/peerj.19859_

## Round 0.1 · original submission · Major Revisions

Dear Authors

Three experts in the study field have reviewed the manuscript. Comprehensive comments to improve the quality of the manuscript have been addressed. I would encourage you to address the sample size estimation and methods reproducibility as the points raised by the reviewers. We invite you to submit a revised manuscript version addressing the reviewers’ comments.

We look forward to receiving your revised manuscript.

Best regards

Yung-Sheng Chen, Ph.D.
Academic Editor

·

Basic reporting

The manuscript is written in clear, professional English. The terminology is appropriate for the field, and the flow from introduction through to discussion and conclusion is coherent. Figures, tables, and equations are well-labeled and integrated logically within the text.
The introduction provides solid context regarding motor impairments, limitations of traditional assessment tools, and the advantages of combining kinematic and sEMG data. Citations are appropriate and recent, including foundational studies (e.g., Falconer & Da, 1985) and current literature (e.g., Maura et al., 2023).

Figures, Tables, Raw Data:
• Figures are relevant and high quality.
• Table 1 appropriately summarizes ROM and angular velocity data.
• Tables 2 and 3 present statistical results clearly.
• Raw data were referenced and described well; assuming it's made available per PeerJ’s policy, this criterion is met.
The study is self-contained, with a well-defined hypothesis (establishing normative benchmarks for upper limb control) and corresponding results that support it.

Experimental design

Sample size justification: please justify 20 subjects is enough for this study. I don’t think it’s enough as this study aimed to establish a normative. Additionally, why just recruiting 20 healthy subjects? The goal of this study was more clinical populations, right? The norm healthy general subjects makes this study less interesting.

Methods Reproducibility:
The methods, including instrumentation, data acquisition, signal processing, segmentation, and statistical analysis, need to be more detailed to allow reproduction. The current form, unfortunately, is not able to reproduce. I suggest adding one or two flowcharts.

Please specify the statistical methods you used, and then in the Results section, present the results.

Validity of the findings

Unfortunately, I don't see the results were presented the way they should be.

I will provide more comments once you have a comprehensive result.

Reviewer 2 ·

Basic reporting

The article was written with appropriate background and literature review. Professional structure format has been provided with precise figures, tables and available raw data.

Experimental design

Research question was well defined and clinically meaningful. The methods were described with sufficient detail to be replicated by others. The experimental protocol has been well established to evaluate joint mobility comprehensively.

Validity of the findings

This study investigated motor control during elbow FE movements in healthy adults, aiming to establish normative biomechanical and neurological benchmarks. In actual daily living, upper extremity involves not only elbow flexion-extension but three-dimensional movement across multiple joints.
In order to expand the dataset to the population with motor impairment, muscle coactivation should be investigated during task-oriented synergistic movement.

Additional comments

As declared by the authors, multidimensional approach to upper extremity is needed. The rotational movement across multiple axis should be assessed for further clinical translation.

Reviewer 3 ·

Basic reporting

The manuscript is well written, the aim of the study is well defined as well as the experimental protocol. The Figures should be exported in higher quality.
The major issue is on the statistical analysis:

1)ANOVA results should be reported somewhere (they are not in the tables). What is effect of the factors (speed and movement phase) in the CC? Are they independent or is there a factor effect between the two?

2) The analysis is very confusing and hard to interpret with respect to the results in the previous section. for Table2: Did the authors pool together all "CC_for_phase_movement" results, regardless the speed, and compared them (while doing the opposite for Table3)? If so, the reader would be facilitated by providing boxcharts of these distributions, on which (*) can be drawn to highlight statistical difference between groups.

3) The significance level of 0.05 is erroneous when performing multiple comparison, Multiple comparisons inflate the risk of false positives (Type I errors), meaning that with more tests, the chance of incorrectly declaring a significant finding increases. Significance level corrections (like Bonferroni or FDR) adjust the criteria for significance to maintain the overall error rate when many tests are conducted. These corrections help ensure that the conclusions drawn from the data are robust and not artifacts of random chance due to multiple testing.

4) It is not clear to me the hypothesis moved prior to the statistical tests. What do the Authors want to assess with them?

5) No significance was found for groups related to speed factor. Looking back at the previous results this is how I would suggest to re-organize this subsection:
a) First present the ANOVA statistical analysis results (showing the graphical distribution of these two groups);
b) As no effect was observed due to speed of the movement, the CC can be pooled together in terms of movement phase, regardless the speed
c) Perform pair-wise post-hoc test to assess the difference between CC_movement_phase groups and report the quantitative results in terms of average \pm std

6) In my opinion for all results it would be better to show boxcharts rather than barplots with std, as the formers allow to visualize the distribution of the quantities

Experimental design

The experimental protocol is well defined with sufficient details and information to replicate. However, the Authors should clearly state the hypotheses moved prior to the design of the protocol. This will help in better understanding the research question and the reason behind specific analyses conducted.

Validity of the findings

While I understand the reasoning behind this study, I believe the findings lack novelty as most of the results were already observed in literature (As also the Authors cited in the Discussion).

An additional effort could be done to improve the manuscript quality and relevance by providing a preliminary comparison of the metrics with results from literature on the target population (i.e. neurological disorders). While I am aware that the experimental settings might be different, the elbow flex extension is a very standard movement and I believe some assumptions could be drawn regardless (One of the main issues in the "assessment field" is the lack of proper validation of existing methods, as novel metrics and experimental protocols are often introduced instead). One reference example could be the following study of Dr. Levin, in which some quantities were calculated from elbow flexor and extensor muscles in post-stroke survivors:

Levin, M. F., Selles, R. W., Verheul, M. H., & Meijer, O. G. (2000). Deficits in the coordination of agonist and antagonist muscles in stroke patients: implications for normal motor control. Brain research, 853(2), 352-369.

Annotated reviews are not available for download in order to protect the identity of reviewers who chose to remain anonymous.

---

## Round 0.2 · Minor Revisions

Dear Authors

Three experts in the study field have reviewed the manuscript. Still, several comments should be addressed before acceptance for publication. I would encourage you to address the method for effect size and the figure legend/resolution as the points raised by the reviewers. We invite you to submit a revised manuscript version addressing the reviewers’ comments.

We look forward to receiving your revised manuscript.

Best regards

Yung-Sheng Chen, Ph.D.
Academic Editor

·

Basic reporting

I would like to thank the authors for addressing my comments and revising the manuscript. The revised manuscript has improved the paper significantly. I have a few minor suggestions, which I will provide in the following paragraphs.

As for Figure 1, I appreciate the authors adding the figure, and I am aware that the description of the contents of the figure can be found in the Methods section. The figure needs to be self-explanatory with the figure legend. Therefore, I believe more information should be added to the figure legend. For example, instead of listing "mDurance sensor" "3 gel surface electrodes"..., the authors can list the variables in the boxes, and add information if necessary.

Line 104, 29.69 +/- 8.01 years (don't forget the unit). Also, can other subject characteristic information be listed (e.g., height, body weight)?

Line 224, please double check the effect size reference. "Cohen’s f statistic was also calculated to assess the strength of the statistical differences", please rephrase this to "assess the magnitude of the effect...".

Experimental design

None

Validity of the findings

None

Additional comments

None

Reviewer 2 ·

Basic reporting

no comment

Experimental design

no comment

Validity of the findings

The reviewer agreed that a single-plane well-controlled task is a necessary prerequisite before applying the baseline measurement to more complex movements or clinical situations.

Additional comments

no comment

Reviewer 3 ·

Basic reporting

The authors implemented most of the required changes. The figures are still in low quality and exporting them with >300dpi is recommended.
The novelty of the work is still questionable.

Experimental design

no comment

Validity of the findings

the findings still represent a little contribution to the state-of-the-art.

Additional comments

no comment

---

## Round 0.3 · accepted · Accept

Dear Authors

Your submission is now endorsed for acceptance of publication in PeerJ. Thank you for submitting your article to PeerJ. I would like to express my gratitude for your contributions and efforts to the scientific community. I look forward to receiving your research and review articles in the future.

Best Regards


Yung-Sheng Chen, Ph.D.
Academic Editor

·

Basic reporting

Thank you for addressing my last comments. I believe the manuscript is ready to go!

Experimental design

None

Validity of the findings

None

Additional comments

None